# Feature Reduction for the Classification of Bruise Damage to Apple Fruit Using a Contactless FT-NIR Spectroscopy with Machine Learning

**DOI:** 10.3390/foods12010210

**Published:** 2023-01-03

**Authors:** Jean Frederic Isingizwe Nturambirwe, Eslam A. Hussein, Mattia Vaccari, Christopher Thron, Willem Jacobus Perold, Umezuruike Linus Opara

**Affiliations:** 1Eresearch Office, DVC Research and Innovation, University of the Western Cape, Private Bag X17, Bellville 7535, South Africa; 2Inter-University Institute for Data Intensive Astronomy, Department of Physics and Astronomy, University of the Western Cape, Bellville 7535, South Africa; 3Inter-University Institute for Data Intensive Astronomy, Department of Astronomy, University of Cape Town, Rondebosch 7701, South Africa; 4Department of Science and Mathematics, Texas A&M University-Central Texas, Killeen, TX 76549, USA; 5Department of Electrical and Electronic Engineering, Stellenbosch University, Private Bag X1, Matieland 7602, South Africa; 6SARChI Postharvest Technology Research Laboratory, Africa Institute for Postharvest Technology, Faculty of AgriSciences, Stellenbosch University, Private Bag X1, Matieland 7602, South Africa; 7UNESCO International Centre for Biotechnology, Nsukka 410001, Nigeria

**Keywords:** variable selection, model optimisation, defect classification, machine learning, baseline, uncertainty quantification, feature reduction, quality control, bruise damage, apples

## Abstract

Spectroscopy data are useful for modelling biological systems such as predicting quality parameters of horticultural products. However, using the wide spectrum of wavelengths is not practical in a production setting. Such data are of high dimensional nature and they tend to result in complex models that are not easily understood. Furthermore, collinearity between different wavelengths dictates that some of the data variables are redundant and may even contribute noise. The use of variable selection methods is one efficient way to obtain an optimal model, andthis was the aim of this work. Taking advantage of a non-contact spectrometer, near infrared spectral data in the range of 800–2500 nm were used to classify bruise damage in three apple cultivars, namely ‘Golden Delicious’, ‘Granny Smith’ and ‘Royal Gala’. Six prominent machine learning classification algorithms were employed, and two variable selection methods were used to determine the most relevant wavelengths for the problem of distinguishing between bruised and non-bruised fruit. The selected wavelengths clustered around 900 nm, 1300 nm, 1500 nm and 1900 nm. The best results were achieved using linear regression and support vector machine based on up to 40 wavelengths: these methods reached precision values in the range of 0.79–0.86, which were all comparable (within error bars) to a classifier based on the entire range of frequencies. The results also provided an open-source based framework that is useful towards the development of multi-spectral applications such as rapid grading of apples based on mechanical damage, and it can also be emulated and applied for other types of defects on fresh produce.

## 1. Introduction

Apple fruit are highly susceptible to mechanical damage resulting from handling practices during and after harvest. Such damage are characterized by tissue and cell deterioration and facilitate infections by microorganisms and disease spread, leading to fruit spoilage and thus postharvest loss. Damage prevention measures that are applicable to handling can help reduce bruise occurrence [1,2,3]. However, such measures are limited by the requirement for personnel with expert training to do the handling, which is not always feasible, especially in developing countries [4]. Grading and sorting of produce based on the presence and degree of defects can help in repurposing them for appropriate uses, such as animal feed or processing when their consumer acceptability is not ideal for market display, and thus reduce the likelihood for further disease spread if fruit skin is broken [5,6]. This can also be an alternative and/or complementary solution for further reduction of losses and ensuring quality and safety of fresh produce.

Non-destructive techniques (NDT) for evaluating the presence of damage on fruit have seen improvements over the years [7], whereby techniques such as optical coherence tomography [8], multispectral imaging [9] and thermal imaging [10] among others, show promise for effective sorting and grading. Multispectral imaging relies on few, fixed wave bands that are most descriptive of a target defect or quality parameter and has the advantage that it can enable fast detection at industrial sorting speeds [11]. However, for each application the determination of specific wavebands that are the best determinant of the properties that are relevant to the application is required [12]. Vibrational spectroscopy is also a prominent option for defect detection, but produces high dimensional spectral data where many variables may contain information that is irrelevant to the problem at hand. The use of full spectra results in models that are complex and of which performance may be impaired by the inclusion of less informative variables. An efficient way of optimizing models in terms of simplicity and performance aims at selecting and including only variables that are most informative to the model [13].

There are various variable selection methods, but none is fit for all purposes [14]. However, studies on detecting various defects have reported successful model improvements using variable selection [15,16]. Using weighing coefficients of the best PC images, Huang et al. (2015) proposed 780, 850 and 960 nm as effective wavelengths within the range of 325–1100 nm, for detecting bruises on apples as an attempt to develop a multispectral imaging (MSI) system for online use. Nturambirwe et al. (2018) found that GA-PLS was consistently improving full spectra-based bruise classification models by a margin from 10% to 30% in terms of classification accuracy, when using a contact mode and 22 mm spot sample size scanning [17].

The enabling factors for industry applications include instrumental designs that are suitable for industrial systems such as capability for large sample exposure and fast scans for data acquisition, robust detection models and the ease of calibration transfers [18], as well as open platforms for collaborative development efforts. Other uses of NDT for food quality and material identification by various users, such as consumers and researchers have also known a growing interest [19] and favors user-friendly handheld and mobile spectrometer designs. The effectiveness of these applications rely on the proper identification of application specific and relevant wavelengths [20,21].

In this work, the aim was to determine the significant wavelengths for bruise discrimination in three apple cultivars, using a Fourier Transform Near-Infrared (FT-NIR) spectrometer that simulates online sample presentation (contactless exposure of large sample size up to 100 mm in diameter). The objectives were to first develop an open source software-based machine learning pipeline for modelling FT-NIR spectral data, secondly, establish the importance of wavelengths as it relates to bruise damage in apples, and lastly, provide a context of application prospects.

## 2. Materials and Methods

### 2.1. Fruit Material

Three apple cultivars, namely ‘Golden Delicious’ (GD) (yellowish green), ‘Granny Smith’ (GS) (green) and ‘Royal Gala’ (RG) (predominantly red), were acquired in two installments (in two consecutive months) from two different local retail shops from Stellenbosch, Western Cape, South Africa, in 2019. A batch of 100 apples were sourced first (source S1) and 114 apples were acquired in the second instance (source S2) with nearly equal proportions of cultivars. Fruits that were free from visible defects were selected and used in the bruising experiment.

### 2.2. Experimental

The apples were kept in cold storage (5 ${}^{\circ }$C, 85% RH) pending a bruising experiment and Fourier transform (FT)-NIR spectral measurements thereafter. They were left at room temperature for three hours prior to each bruising experiment, in order to carry out measurements at ambient, laboratory conditions (25 ${}^{\circ }$C, 65% RH). Bruise damage was created by dropping a stainless steel ball from different heights (20, 35 and 65 cm) on two opposite sides of each apple, thus creating bruises with three degrees of severity. Experimental setup was done according to [6,22]. Two areas on opposite sides (bruised and non-bruised) around the equatorial plane of every apple were scanned under the non-contact emission head (EH) of the Matrix-F spectrometer (Matrix-F duplex from Bruker Optics, Ettlingen, Germany). For each single measurement, the spectrum was averaged over 64 scans. The NIR scanning range was between 12,500–4000 cm${}^{-1}$, in intervals of 4 cm${}^{-1}$ [23]. The MATRIX-F FT-NIR spectrometer is equipped with a fiber optic NIR illumination and detection head (185 mm height and 230 mm diameter for sample sizes up to 100 mm in diameter) and allowed for measurement on half of the whole apple surface per single exposure. The fiber optic illumination head contains 4 air cooled tungsten NIR light sources (Tungsten halogen, 12 V, 20 W). The diffusely reflected light from the sample was collected and guided via a fiber optic cable to the spectrometer detector (a highly sensitive, thermoelectric cooled and temperature controlled InGaAs diode detector) [24].

### 2.3. Sampling

From the three apple cultivars, two main sample categories were created, namely bruised (B) and non-bruised (S) fruit. From the B category, three subcategories were created by representing the different levels of bruise severity (L1, L2, and L3), thus contributing to more variability in the data set.

The ratio between the two classes indicates that the data are balanced. Figure 1 shows the 3 apple data sets with 50 randomly selected infrared samples.

### 2.4. Data Pre-Processing

Various spectral preprocessing methods including Multiplicative Scatter Correction (MSC), Standard Normal Variate (SNV), derivatives, scaling and normalization can enhance the modeling outcome in spectral signals [25] data and were applied to the spectral signals. “Standard scaling” was found to be a most viable option. All three data sets were standardized using the StandardScaler library implemented in scikit-learn [26] and the Python programming language.

Figure 2 shows the standardized wavelength data for the three different apple types. Visual comparison with Figure 1, shows that standardization produces a clearer separation between B and S apples, especially for GS and RG apples.

### 2.5. Analytical Workflow

#### 2.5.1. Introduction

Our main goal in this investigation is to reduce the number of features, thus simplifying the classification process. In this section, we describe the workflow used to select optimal features and machine learning methods for bruise classification. The code and the data together with the results, are available on Zenodo [27].

#### 2.5.2. Baseline Method

We also established logistic regression (LR) on all 2000 wavelengths as a baseline predictor against which all other predictors can be compared. As as a preliminary test, we measured feature permutation importance on all wavelengths using the permutation_importance function from scikit_learn. The permutation importance test measures the relative drop in accuracy when each individual feature is shuffled, thus destroying the correlation between that feature and samples [26,28]. This method has the advantage of fast execution (which is necessary since we are examining more than 2000 features). Figure 3 shows the relative importance of the first 500 features. The figure shows that there are clusters of wavelengths with higher feature importance. However, these wavelengths are supplying redundant information, since the intensity values vary only slightly between adjacent wavelengths.Thus we expect that many wavelengths can be eliminated as features without greatly reducing the predictive accuracy.

#### 2.5.3. Feature Selection

The permutation method clearly shows that some wavelengths are more important than others. However, the permutation method itself is unsuitable for feature selection, since it does not take into account the fact that two different relatively important features may be highly correlated and supply similar information.

To deal with selection among correlated features, there are two commonly-used methods, described as follows:In recursive feature elimination (RFE), the model is first trained on the entire set of features, and deletes the least important features. This training-deletion procedure is repeated recursively on the remaining features until the desired number of features remain.In sequential feature selection (SFS), initially models based on each individual feature are computed, and the feature with the best cross-validation score is selected. Then all models consisting of the selected features and one additional feature are computed, and the best chosen. This process is repeated recursively, adding one feature each time. The default cross validation score was used for this step. There is a backward selection variant of this method that is similar to RFS but we found that execution is too slow.

#### 2.5.4. Machine Learning Classifiers

Mathematical models have been used extensively for chemometric studies of spectral data from non-contact FT-NIR acquired on fruit. Machine learning models are increasingly being used for this purpose [6,29] because of their flexibility and adaptibility to a wide variety of applications. A total of six machine learning (ML) tools are used for the binary classification: logistic regression (LR), support vector machine (SVM), random forest (RF), extreme gradient boosting (XGB), *k*-nearest neighbour (Knns), and Artificial neural networks (ANN). These methods are state-of-the-art in the literature for relatively small datasets [30,31] such as those described in Table 1.

All tools were optimized with 3-fold cross validation to avoid overfitting to make sure that a trained model can generalize on unseen data.

Optimization of ML parameters for all feature sets was performed used random search, as implemented in the Python library scikit-learn [26]. All models except for SVM and the baseline went through 30 randomized searches, whereas SVM went through only 15 because SVM is very computationally intensive.

#### 2.5.5. Comparison of Classifier Performance

In this research, we compared the classification precision of all ML methods for all features, which is computed as
(1)$$\mathrm{Precision}=\frac{TP}{TP+FP}$$
where *TP* and *FP* are the numbers of true positive and false positive predictions, respectively. Here a ‘positive’ result refers to identifying the apple as good: so a *FP* result falsely classifies a bruised apple as good. Precision is used because the most costly error is misidentifying bruised apples.

To make an effective comparison between models, error bars (corresponding to two standard deviations) for the classification precision differences were also calculated. If 0 lay outside the error bars, then we concluded that the difference between the two models was statistically significant. Otherwise, we failed to reject the null hypothesis of no difference between models.

The precision of the different classifiers were evaluated using the testing set. In order to obtain error bars, jackknife with leave-out-one was implemented [32,33]. For the training set error, the left-out instances were 1/3 of the training set, while for the testing set all instances were left out one-by-one to obtain the jacknife estimate of the standard deviation.

## 3. Results

In our workflow, we used both RFE and SFS methods with LR in order to get the best 50 features, which resulted in 50 × 2 classifiers, corresponding to taking the best feature, best 2 features, up to the best 50 features. The choice of 50 features was based on preliminary investigations using permutation importance, which showed that no gain in accuracy was achieved from using more than 50 features. Figure 4 shows the precision score for the best 50 feature sets using LR on training data. The figure shows that SFS gives better estimators for all three apples for all features sets. Consequently, for the testing results we only applied SFS and did not consider RFE. Note that the feature selection in Figure 4 employed all 2000+ features, which was a very computationally intensive calculation. Subsequently we found that virtually identical results could be obtained with much less computation time by doing feature selection with only the best 200 features obtained from permutation importance (shown in Figure 3).

Figure 5 shows graphically the wavelengths for the best 10, 20, 30, 40, and 50 features for all the three cultivars: the exact wavelength values are given in Appendix A. The *x* axis on each plot gives the wavelength, while the *y* axis shows the feature ranking in groups of 10. The figure shows that the 10 most important features are somewhat consistent among all three cultivars: wavelengths near 900, 1300, 1500 and 1900 nm appear as top-10 wavelengths for all three cultivars. Nonetheless, there are significant differences between the best features for the three cultivars. For example, only GS has wavelengths above 2400 in the top 10. It should be noted that in an industrial setting cultivars would be sorted separately, hence in practice a cultivar-specific approach to feature selection is desirable. We also note that as feature importance decreases, the features tend to get more clustered. For example, with GD apples the features with importance 41–50 form three clusters, while features with importance 0–30 are more diverse.

From the 50 different features sets we selectedP four for testing on the testing set. The four features sets were determined as follows: (a) Features up to and including the first jump in accuracy; (b) Best 10 features; (c) When the model stabilizes; (d) Best 50 features. Figure 6 compares the precision of models based on the four different feature sets. For each feature set, all six ML tools listed in Section 2.5.4 were implemented. In general precisions ranged from 0.7 to 0.9, with GD typically obtaing the best results. For each ML method feature sets (b)–(d) gave nearly the same performance, LR and SVM obtained similar mean precisions, but LR had much smaller error bars. The overall highest mean precisions for the different apple species were 0.86, 0.79, and 0.81 for GD, GS, and RG respectively. All of these best results were obtained with feature set (c).

The horizontal line in Figure 6 shows the precision obtained with the baseline estimator which was LR using all wavelengths. Compared to the baseline, the best-performing estimators were slightly lower mean precisions, but differences were small (between 0.01–0.03) and statistically insignificant. Although the reduced feature sets did not give better precision than the baseline, they require far fewer wavelength measurements and correspondingly are much easier to implement in practice.

## 4. Discussion

The precision of classification for all models based on selected features subsets developed in this study did not exceed that of the baseline, which utilized all the wavelengths. A similar relative performance was also reported in a study to classify bruises in four apple cultivars by Luo et al. [16]; authors applied a ROC-AUC based method to select effective wavelengths for spectral imaging and found that reduced models did not exceed the performance expressed by sensitivity, specificity and accuracy of full-spectrum based model. It should be noted that though our binary classification focuses only on the existence or absence of bruises, our sampling has introduced high variability in the sample space by including different levels of bruising, sample origins and sampling times. High variability in the samples can be a challenge for some models to fully capture. However, it is a desirable aspect to build models with high generalisation ability. Furthermore, the reflective mode of spectral acquisition may introduce nonlinear relationships between chemical composition and spectra [34], as opposed to the linear relationships indicated by the Beer-Lambert law.

In all the cultivars, the LR and SVM models based on about 40 wavelengths had the highest average precision values (0.79–0.86). These results are comparable to the common range of classification performance metrics in spectroscopy-based bruise classification. Nonetheless, there has been reports [35] of higher model performance for bruise classification in apples using different absorption wavelength regions. There is some possibility that higher performance may be obtained using deep learning, which has been applied successfully in other agro-product classification scenarios [36]. However, effective training of a deep learning classifier typically requires a much larger dataset than was available for this study [32,37]. Furthermore, many studies that apply machine learning (including deep learning) do not include error bars in their performance evaluations, so there is some question as to whether reported improvements in classification are statistically significant [38].

The generalization ability of machine learning is an important aspect for effective applications. Therefore, all the models were tested on unseen data, previously separated from the original dataset. Nonetheless, given that data were acquired in a controlled manner, it should be noted that factors such as temperature may affect the acquisition of NIR spectra. Also, variability can be introduced by considering fruits from different climatic regions and growing conditions. To improve model generalization ability, these aspects could be considered in future studies.

Previous studies have examined the NIR waveband characteristics of bruises in apples. Geola et al. [39] proposed a procedure for detection of damaged tissue in ‘Golden Delicious’ apples, and a classification function was obtained in the region of 750 to 800 nm. Other apple tissue classification studies have used reflectance spectra, and in most cases the most significant wavelengths were found in the range of 690–850 nm [40,41,42]. Kleynen et al. [43], working with a VIS-NIR spectrometer developed a method to detect defects on bicolor apple fruit (‘Jonagold’). They found that the most significant wavelengths were in the NIR range (700–920 nm or 14,285–10,869 cm${}^{-1}$). According to Lammertyn and associates, the light penetration depth of NIR radiation in Jonagold apple tissue was the highest in the region 700–900 nm [44]. Hence, Kleyman associated their observations to the latter wavelength region. Although these studies were generally based on short-wave NIR bands, considering the broad spectrum range from 780 nm to 5000 nm can improve detection of bruises with varying depths [10].

The chemistry of apple peel is very complex and has been adapted for its biological function. The peel’s outer surface consists of cuticle waxes composed of fatty alcohol, fatty acids and long chain hydrocarbons [45,46,47]. Additionally, there are different class of compounds such as flavonoids [48,49], phenolic acids [50,51], triterpene esters [52] and proanthocyanidins [53]. The color chemistry of the peel varies according to the cultivar of the apple under consideration; the major class of compounds contributing to the apple’s colors are chlorophylls (green) [54], carotenoids (yellow) [55] and anthocyanins (Red) [56].

The NIR spectrum has several overtones due to absorption from several different types of chemical bonds including C–H (from methyl, methylene, methoxy, aromatic, and carbonyl associated groups, N–H (from primary and secondary amides, primary, secondary, and tertiary amines, and amine salts) O–H (alcohols and water), S–H, and C=O groups [57].

In our case, it is difficult to localize any specific signals match with any of the functional group; from Figure 1, several broad bands can be identified which are common to all three apple types. The first band ranges from 850-1100 nmm and includes two sub-bands: 850–920 nm (C-H methyl/methylene associated with aliphatic and/or aromatic skeletons and 950–1000 nm (OH and NH aliphatic /aromatic groups), A second band ranges from 1150–1300 nm is due to C=O, aromatic C–H and C=C groups, A third band ranges from 1350–1600 nm, and covers OH, free and hydrogen bonded and C–H aliphatics and aromatics, N–H aromatics and aliphatics, C=O aldehydes and ketones, A less prominent band ranges from 1800–1850 nm, and covers OH, CH of carbohydrate polymer. Finally, a band from 1900-2000 covers C=O amides/carboxylic groups, and OH/NH mainly aromatics.

Figure 7 shows how the most important features for bruise identification compare with the NIR spectra for the different apple types. In every case there are top-10 features associated with each band in the spectrum. However, the selected features tend to be located near the steep rising edge of the band, and not in the middle of the band range. The band associated with the most top-10 features is 1350-1600, which as described above includes absorption from several types of chemical bonds. Exact values of the top-10 features/wavelengths for each apple type are given in Appendix A.

Applications of spectroscopy to detect bruises seem to have had diminished interest in favor of imaging options. This may be related to the spatial limitations of spectroscopic devices, which offer only a limited spatial representation (up to 22 mm spot diameter) of the sample exposed. This limitation makes it inconvenient for most uses for fast sorting and grading of different types of fresh produce. The latest trends in non-destructive defect detection in agri-food fresh products have been focused on imaging techniques [58], including hyperspectral imaging. However, these imaging techniques rely heavily on image processing for feature extraction which is computationally costly. The emerging use of deep learning for spectroscopy and imaging based evaluation of agro-product quality offers a quick end-to-end modelling process [36], but the acquisition of images is still time consuming and not fit for industrial sorting speeds. Hence, multi-spectral imaging is preferred for such applications, since it is based on a few predetermined wavelengths that are effective for a specific attribute evaluation. This work shows the feasibility, with the flexibility of open-tools software, of wavelength selection on spectral data from fully exposed apples. This kind of exposure simulates sample presentation that would be applicable for inline sorting applications.

## 5. Conclusions

Building on the possibilities offered by a combination of open-source development tools and a contactless NIR spectrometer with large (100 mm in diameter) sample exposure, a feasibility study of bruise damage classification in apples was conducted, with the aim to determine the informative wavelengths in three apple cultivars. Bruise segregation models were built using six machine learning classification algorithms coupled with both, recursive feature elimination and sequential feature selection methods to determine the most informative wavelengths. A classification precision that matched the full-spectra based models within error bars could be achieved using up to 50 wavelengths selected from 4 main wavebands. The best reduced classification models were based on the LR and SVM machine learning techniques, which gave precision values ranging from 0.7 to 0.9 depending on the cultivar.

## Figures and Tables

**Figure 1 foods-12-00210-f001:**
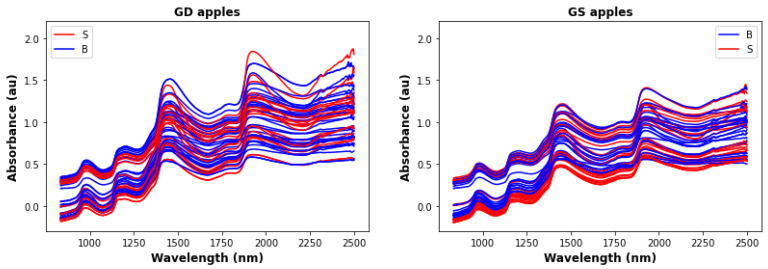
Infrared data for 50 randomly selected GD, GS, and RG, where B are the bad apples, and S are the good apples.

**Figure 2 foods-12-00210-f002:**
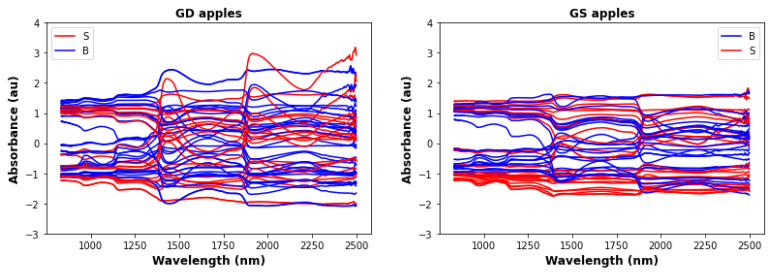
Infrared data after standardization for the same samples as in Figure 1.

**Figure 3 foods-12-00210-f003:**
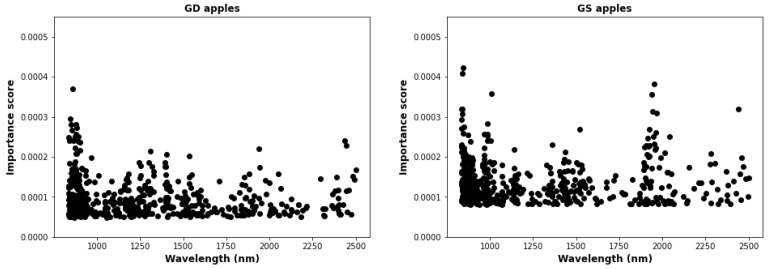
The graph shows the permutation importance for the top 500 features/wavelength.

**Figure 4 foods-12-00210-f004:**
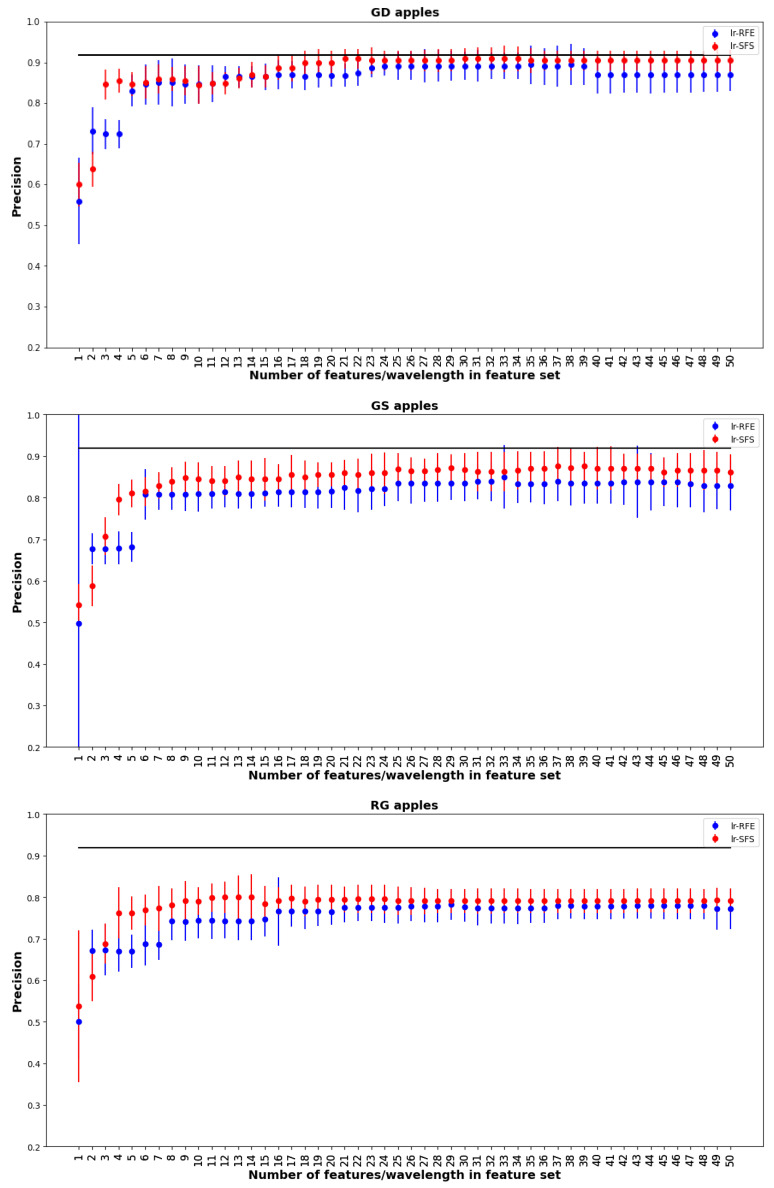
Best 50 feature sets using RFE and SFS on the 3 data sets using LR.

**Figure 5 foods-12-00210-f005:**
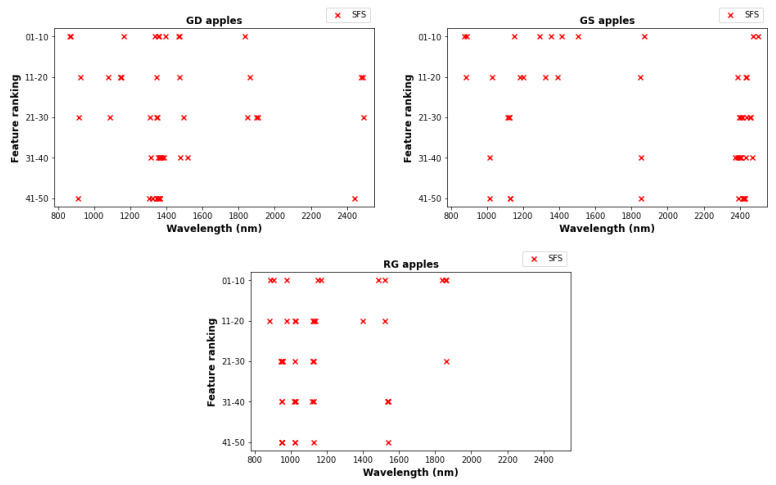
Allocation of the 50 selected features that are most relevant to bruise segregation for the three cultivars. Selected wavelengths are plotted in sets of 10 with the first selected on top. Detailed frequency information is given in Appendix A.

**Figure 6 foods-12-00210-f006:**
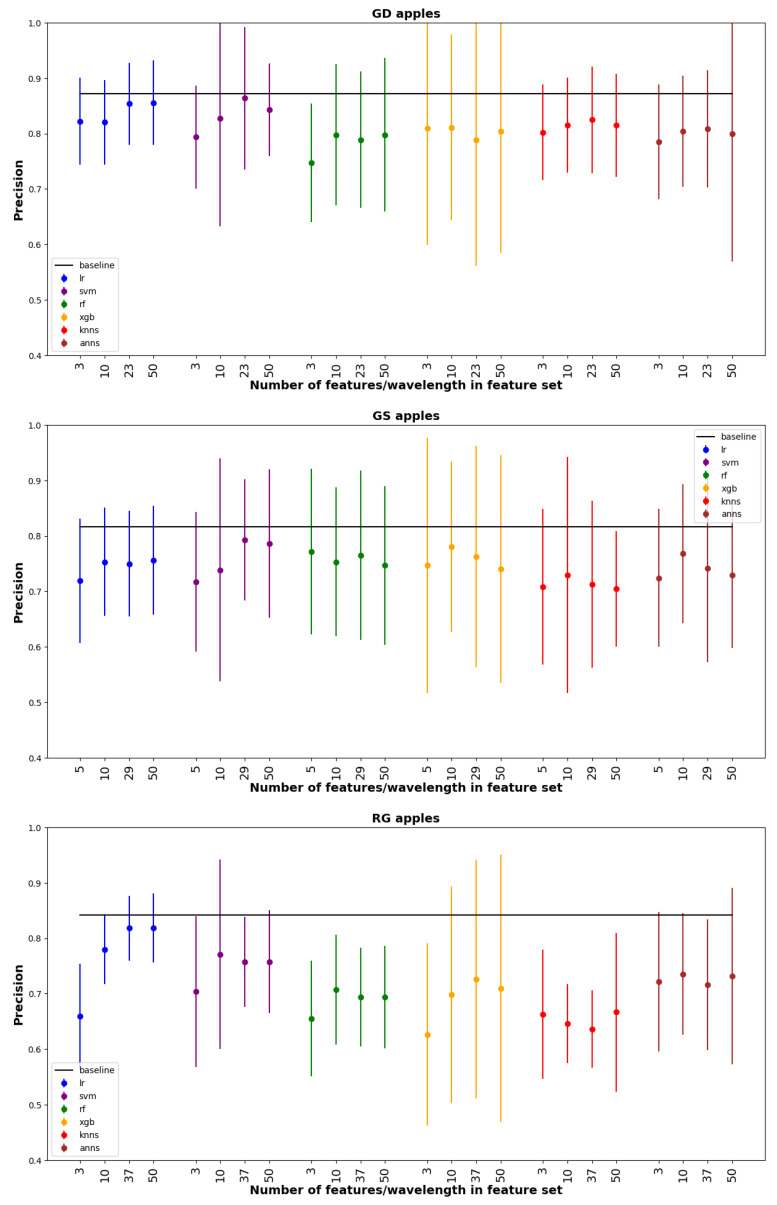
Plot of precision for all models using and all selected feature subsets, on test data with a comparison to the baseline model (horizontal line). The vertical lines represent error bars.

**Figure 7 foods-12-00210-f007:**
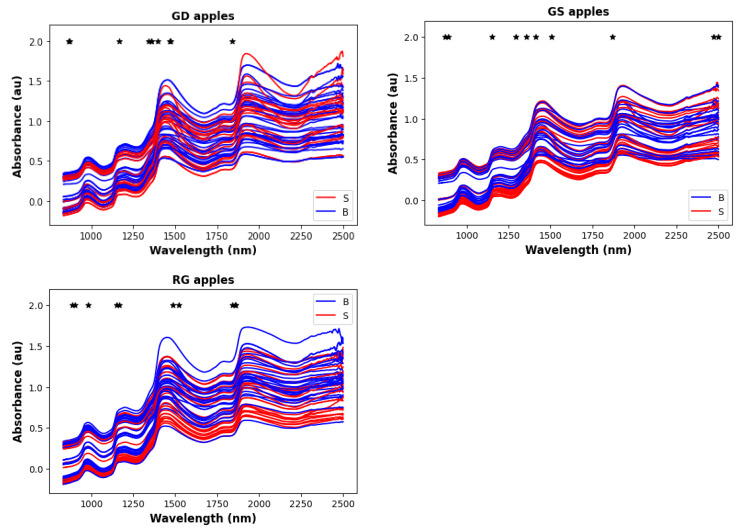
Position of top 10 features/wavelengths (shown as black asterisks) compared to the NIR spectra for the three different apple types.

**Table 1 foods-12-00210-t001:** Repartition of the number of samples per apple cultivar.

Type	Non-Bruised Samples	Bruised Samples	Total
GD	274	273	547
GS	252	251	503
RG	278	284	562

## Data Availability

The code, together with the results, is available on Zenodo at https://zenodo.org/badge/latestdoi/478611734). We also provided a walk-through educational tutorial which can be found on Zenodo at https://doi.org/10.5281/zenodo.7018476.

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
