# Peer review of "Feature Reduction for the Classification of Bruise Damage to Apple Fruit Using a Contactless FT-NIR Spectroscopy with Machine Learning"

_foods, 2023, doi:10.3390/foods12010210_

Round 1

Reviewer 1 Report

Dear authors, 

I read several similar papers like this one, and I must notice that the current manuscript fails to provide all the information that I saw from other studies. Based on what you currently provided, the paper is rather "technical" and as such I can recommend it for major revisions. I would strongly suggest to dig more into the data you have and results you obtained, because the publication should not be only about applicable aspects of the research, but also give scientific value. I did not see much of the scientific contribution, and I would like to see at least the table with the wavelengths you found are important for bruises detection and appropriate assignments of the bands you observed. Near infrared is not "black box" and should not be treated as such. Please find the annotated pdf with my comments and questions and revise the paper accordingly. 

Best regards

Author Response

Dear Reviewer

The authors are grateful for the constructive feedback you have provided which served us in improving our manuscript. We have considered every comment carefully and written a point-by-point response on how we have addressed the feedback. Please, find the response attached.

Reviewer 2 Report

Dear authors,

Your work is very well done. I have not detected substantial methodological problems and the results seem to me to be interpreted correctly.

The only remark I make that in my opinion requires your review is that the arrangement of the manuscript seems to mix the methods with the results. Therefore, I suggest moving Figures 1-3 with their comments to the part of the results, leaving in the methodological part only the strict description of the methods and the reason for their application.

Author Response

Dear Reviewer

The authors appreciate your constructive review feedback. We have considered carefully the recommendations and have addressed them point-by-point as mentioned in the response attached. 

Kind regards
The authors

Round 2

Reviewer 1 Report

The authors have performed all the requested revisions, and I think the paper can now be accepted for publication.